# Glyphosate-Based Herbicide Formulations with Greater Impact on Earthworms and Water Infiltration than Pure Glyphosate

Verena Brandmaier [1], Anna Altmanninger [1], Friedrich Leisch [2], Edith Gruber [1], Eszter Takács [3], Mária Mörtl [3], Szandra Klátyik [3], András Székács [3] and Johann G. Zaller [1,*]

1   Department of Integrative Biology and Biodiversity Research, Institute of Zoology, University of Natural Resources and Life Sciences Vienna (BOKU), Gregor Mendel Straße 33, 1180 Vienna, Austria; verena.brandmaier@gmx.at (V.B.); a.altmanninger@gmx.at (A.A.); edith.gruber@boku.ac.at (E.G.)

2   Department of Landscape, Spatial and Infrastructure Science, Institute of Statistics, University of Natural Resources and Life Sciences Vienna (BOKU), Peter-Jordan-Straße 82, 1190 Vienna, Austria; friedrich.leisch@boku.ac.at

3   Agro-Environmental Research Centre, Institute of Environmental Sciences, Hungarian University of Agriculture and Life Sciences, Herman Ottó út 15, H-1022 Budapest, Hungary; takacs.eszter84@uni-mate.hu (E.T.); mortl.maria@uni-mate.hu (M.M.); klatyik.szandra@uni-mate.hu (S.K.); szekacs.andras@uni-mate.hu (A.S.)

*   Correspondence: johann.zaller@boku.ac.at; Tel.: +43-1-47654-83318

**Abstract:** Glyphosate is the most widely used active ingredient (AI) in thousands of glyphosate-based herbicides (GBHs) worldwide. Short-term impacts of AIs or GBHs on earthworms are well known, but few studies have examined long-term legacy effects >3 months after application. In a pot experiment, we studied both short-term and long-term effects on deep burrowing earthworms (*Lumbricus terrestris*) and soil functions. Therefore, the cover crop *Sinapis alba* was grown in soils with either 3.0% or 4.1% soil organic matter content (SOM) and either sprayed with a GBH (Touchdown Quattro, Roundup PowerFlex, or Roundup LB Plus) or the respective glyphosate AI (diammonium-, potassium-, or isopropylamine-salt) or hand weeded (control). Long-term effects showed increased earthworm activity under GBHs even 4 months after application, but similar activity under AIs and control. Another application of the same treatments 5 months after the previous one also increased earthworm activity under GBHs, especially at high SOM levels. Water infiltration after a simulated heavy rainfall was 50% lower, and leaching was 30% higher under GBH than under AI application or hand weeding. Individual GBHs and AIs varied in their effects and were influenced by SOM and soil moisture. Full disclosure of all ingredients in GBH formulations would be necessary to allow a comprehensive assessment of environmental risks.

**Keywords:** agrochemicals; chemical weed control; conventional agriculture; non-target effects; lumbricidae; soil fauna; ecosystem function; soil ecology

## 1. Introduction

Glyphosate is the most widely used active ingredient (AI) in glyphosate-based herbicides (GBHs) [1]. They are used for preemergent weed control on a wide range of annual and perennial crops, preharvest desiccation [2], cover crop removal [3], and in conjunction with no-tillage practices [4]. In addition to agriculture, GBHs are also used for weed control along roadsides, railroad tracks, and in private gardens [1,5].

Thousands of differently formulated GBHs are in use worldwide, containing about 35–50% of the AI glyphosate with numerous co-formulants [6,7]. Co-formulants are considered inert and are reportedly not involved in the toxicity of herbicides, but they facilitate permeation into plant cells, protect the AI from degradation, increase its solubility and half-life, and enhance its effectivity [7]. The amount and type of co-formulants are usually

guarded as trade secrets [8,9]. Studies found that co-formulants such as ethoxylated adjuvants are several orders of magnitude more cytotoxic than pure AI glyphosate [8,10]. Furthermore, it should be noted that AI glyphosate is not a well-defined chemical compound but rather a chemical family that exists in various forms as glyphosate acid or glyphosate salt [11–13]. Therefore, it is important to assess the effects of different glyphosate AIs and their associated GBHs on non-target organisms, especially since environmental risk assessments usually consider only the effects of AIs but rarely the effects of GBHs.

The fate of glyphosate in soil is influenced by sorption and desorption processes, which depend, among other factors, on soil organic matter (SOM). In general, soils with higher SOM can bind glyphosate longer [1,14]. In soil, glyphosate is mainly degraded by microorganisms [15] and can also stimulate soil microorganisms [16]. Its bioavailability is influenced by soil properties such as SOM [17], but also by pH, salinity, and nutrients [18]. The persistence of glyphosate in soil ranges from a 3- to 500-day half-life [2], and it has been found in soils across Europe [19].

Several studies dealing with terrestrial non-target animals show that GBH has the same or higher toxicity than glyphosate AI [20,21]; these studies included soil bacteria [22], amphibians [23], or springtails [12]. In particular, soil fauna is directly [24] and indirectly [25] affected by glyphosate and GBHs. Glyphosate enters the soil through a variety of pathways, including direct application to the soil surface, translocation with rain and air, root exudates, or decomposition of plant residues [1,26]. Anecic earthworm species are particularly affected by glyphosate because they are surface-active and pull contaminated plant material into the soil [27]. Studies have shown that their activity and reproduction are reduced under glyphosate exposure [28–31]. Earthworms are important organisms of the soil macrofauna in temperate agroecosystems [32,33], accelerating the decomposition of plant litter [34] and making it accessible to microorganisms [35]. Through their burrowing activity, earthworms influence water infiltration after heavy rainfalls [13,36,37]. Although earthworms are often used as surrogate species in environmental risk assessments, these tests do not examine the long-term effects of contaminants, i.e., the effects of legacy applications on the previous crop [38]. Long-term legacy effects of GBH (and other herbicides) on microorganisms were observed in the field even 11 months after application [16,39]. Others found GBH-induced changes in plant defense and species interactions in the subsequent crop [40], and these changes are expected to have ecosystemic and even evolutionary consequences for both microbes and hosts [41].

The objective of the present study was to investigate both the short-term and long-term legacy effects of commercial GBHs (Touchdown Quattro, Roundup PowerFlex, and Roundup LB Plus) and their respective AIs (diammonium-, potassium-, and isopropylamine-salt) on earthworm activity and associated ecosystem functions in a greenhouse pot experiment. To date, there have been few studies investigating the effects of GBHs and AIs on earthworms in different soil types, including effects on ecosystem functions [13,17]. We hypothesized that: (i) AI application would have less impact on earthworm activity than GBH because of additional co-formulants but more than hand weeding; (ii) earthworms in soils with higher SOM content are less affected by GHBs or AIs due to sorption of GBHs or AIs; and (iii) long-term effects after application are unclear due to the wide range of half-lives of different GBHs or AIs. If earthworms are affected by GBHs or AIs, we expected that this would lead to cascading effects on water infiltration, glyphosate leaching through the amount of water leached, and litter decomposition.

## 2. Materials and Methods

This study was conducted as a factorial pot experiment that ran from 3 October to 17 December 2018 (75 days in total) in the research greenhouse of the University of Natural Resources and Life Sciences (BOKU), Vienna, Austria. To examine long-term legacy effects, we used soil from a previous experiment [12,13] that examined the effects of the same GBHs, AIs, and soil types as in this experiment. The previous experiment ran between April and July 2018 with an application of GBHs/AIs on 13 June 2018, which is 112 days

before the start of the current experiment. After completion of the previous experiment, the soils were stored in the original pots in the greenhouse for two summer months and allowed to dry out completely. For the current experiment, the pots were seeded with mustard, earthworms were introduced, and the treatments were carried out as follows.

### 2.1. Experimental Setup

The experimental setup consisted of 70 plastic planting pots (height 23 cm, diameter 31 cm, volume 17.4 L) filled with 17 kg of topsoil from the arable fields of the BOKU research farm in Groß-Enzersdorf, near Vienna. Two different soil types were used: half of the pots were filled with soil containing 4.1% soil organic matter (high SOM); the other half of the pots were filled with soil containing 3.0% SOM (low SOM). Table 1 shows the soil properties: both the phosphorus and potassium contents were higher in the high SOM soil, while their pH value was the same. The soils belong to the Chernozem soil type on calcareous ground. The low SOM soil was farmed conventionally in a regional crop rotation with common pesticide treatments; however, no GBHs were applied within the last three years. The high SOM soil has been managed organically for the last two decades and has not been treated with GBHs or other synthetic pesticides ever since.

**Table 1.** Characteristics of the two soils used in the experiment. The analysis and assessment of fertilization levels were conducted by the Austrian Agency for Health and Food Safety (AGES).

| Variable | Soil 1 | Soil 2 | Assessment Soil 1 | Assessment Soil 2 |
|:---:|:---:|:---:|:---:|:---:|
| SOM content [%] | 3.0 | 4.1 | humous | humous |
| Phosphorus [mg kg$^{-1}$] | 73 | 113 | sufficient | high |
| Potassium [mg kg$^{-1}$] | 140 | 234 | sufficient | high |
| pH [CaCl$_2$] | 7.7 | 7.7 | alkaline | alkaline |

A three-factorial design was set up:

- Factor GBH (three levels): one-time application of Touchdown Quattro (TQ), Roundup PowerFlex (PF), or Roundup LB Plus (LB) at recommended dosages (Table 2).
- Factor AI (three levels): one-time application of diammonium salt (am), potassium salt (po), or isopropylamine salt (is) at recommended dosages (Table 2).
- Control: mechanical weeding (co) by pulling plants.
- Factor SOM (two levels): low (3.0% SOM) or high (4.1% SOM).

**Table 2.** Recommended and applied herbicide amounts of the GBHs Touchdown Quattro (TQ), Roundup PowerFlex (PF), and Roundup LB Plus (LB), and their AIs diammonium salt (am), potassium salt (po), and isopropylamine salt (is). Dosage per pot was calculated based on the pot surface of 0.075 m$^2$.

| GBH | AI | Recommended Dosage (L ha$^{-1}$) | AI Content (g L$^{-1}$) | GBH (mL pot$^{-1}$) | AI (g pot$^{-1}$) |
|:---:|:---:|:---:|:---:|:---:|:---:|
| TQ | am | 5.0 | 360 | 0.0375 | 0.0135 |
| PF | po | 3.75 | 588 | 0.0281 | 0.0165 |
| LB | is | 5.0 | 360 | 0.0375 | 0.0135 |

Each factor-level combination was replicated five times, resulting in ((3 GBH + 3 AIs + 1 control) × 2 SOMs) × 5 replications = 70 pots.

### 2.2. Planting and Earthworms

*Sinapis alba* was sown as a model crop because it is a typical winter cover crop in the study region that could be sprayed with GBHs in the spring if mild winter temperatures do not kill this cover crop by frost. Organic-quality seeds were obtained from the company

Reinsaat (St. Leonhard am Hornerwald, Austria). The recommended seeding density is 2–3 g m$^{-2}$ with a 1000-seed weight of 6.27 g. Accordingly, a mean seeding density of 2.5 g m$^{-2}$ or 28 seeds pot$^{-1}$, was sown for the experiment on 11 October. Seeds were buried at a depth of 2 cm in four parallel rows in a standardized pattern (Figure 1). The pots were regularly irrigated with tap water. During the 76 days of the experiment, 5.3 L of water were given per pot.

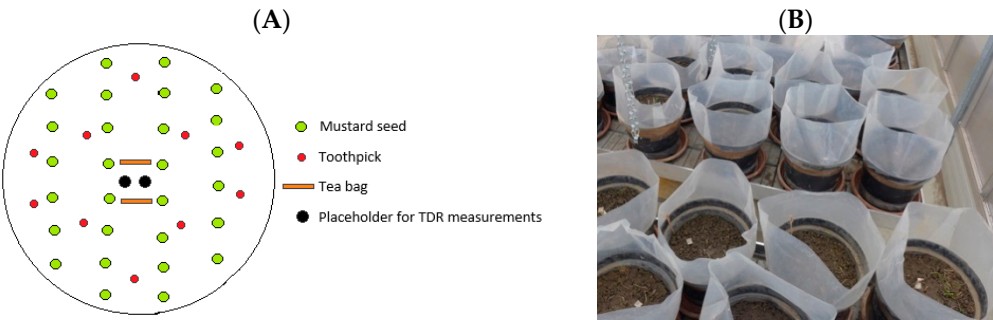

**Figure 1.** Seeding pattern of *Sinapis alba* for each experimental pot; toothpicks to assess earthworm activity; tea bags to measure litter decomposition; placeholders for time-domain reflectometry (TDR) probes to measure soil temperature, moisture, and electrical conductivity (**A**); and final setup of the pots in the greenhouse with earthworm barriers installed on the top (**B**).

On 5 October 2018, three adult earthworms of the vertically burrowing species *Lumbricus terrestris* obtained from a fishing shop (Anglertreff, Vienna, Austria) were introduced into each pot. First, earthworms were rinsed in water, dried with paper towels, and weighed, and 12.1 g ± 2.2 g (mean fresh mass ± SD) were added to the pots. Within a few minutes, the earthworms burrowed into the soil. During the course of the experiment, earthworms were fed with 2 g pot$^{-1}$ of dried, chopped organic hay spread on the soil surface at each feeding date. The hay was purchased from a pet shop (Fressnapf, Vienna, Austria).

To prevent the earthworms from escaping, the bottoms of the pots were covered with a mosquito net, which was attached with adhesive tape. Further, a 25-centimeter-high barrier of transparent plastic film was attached to the upper edge of the experimental units; the upper inner edges of the plastic film were coated with soft soap [31]. The final setup for all pots is shown in Figure 1.

### 2.3. Herbicide Applications

Treatments were imposed on 13 November 2018, 41 days after the start of the experiment and 33 days after seeding. Mustard plants were about 10 cm tall at the time of treatment.

- Factor GBH: consisted of a one-time application of Touchdown Quattro (TQ; AI 360 g L$^{-1}$; Syngenta Agro GmbH; Vienna, Austria), Roundup PowerFlex (PF; AI 588 g L$^{-1}$), or Roundup LB Plus (LB; AI 360 g L$^{-1}$; both Bayer Agrar Austria; Vienna, Austria).
- Factor AI: consisted of a one-time application of potassium salt (AI in Roundup PowerFlex), isopropylamine salt (AI in Roundup LB Plus), or diammonium salt (AI in Touchdown Quattro).
- Control: pots were sprayed with tap water, and plants were uprooted and left on the soil surface of the pots.

GBH's Roundup PowerFlex and Roundup LB Plus were purchased at Raiffeisen-Lagerhaus GmbH (Bruck an der Leitha, Austria); Touchdown Quattro was ordered online from vmd-drogerie (Veseli nad Moravou, Czech Republic).

Glyphosate isopropylamine salt was purchased from Toronto Research Chemicals (North York, ON, Canada), while potassium and diammonium salts were prepared at the

Agro-Environmental Research Institute of the National Agricultural Research and Innovation Centre (Budapest, Hungary) from glyphosate purchased from Sigma-Aldrich, Hungary (Budapest, Hungary). Thus, 1.66 g (9.82 mmol) of glyphosate was gradually added under continuous stirring to a cooled 0.84 mL aliquot of a 45% (wt wt$^{-1}$) aqueous potassium hydroxide solution. The mixture was stirred overnight at 4 °C, and the resultant precipitation was filtered and lyophilized to yield 1.04 g (5.02 mmol, 51.1%) of glyphosate potassium salt. Similarly, 1.66 g (9.82 mmol) of glyphosate was gradually added under continuous stirring to a cooled 1.33 mL aliquot of a 28% (wt wt$^{-1}$) aqueous diammonium hydroxide solution. The mixture was stirred overnight at 4 °C, and the resultant precipitation was filtered and lyophilized to yield 1.01 g (4.97 mmol, 50.6%) of glyphosate diammonium salt.

*2.4. Measurements*

2.4.1. Plant Growth

The height of the mustard plants was measured by hand immediately prior to GHB/AI application and weeding. For this, five randomly selected plants per pot were measured with a ruler from the soil surface to the top of the plant and averaged per pot. The man plant height was 9.8 ± 2.1 cm across all treatments, with no significant difference between treatments.

2.4.2. Earthworm Activity

The surface activity of earthworms was measured using the toothpick method and by collecting earthworm casts [31]. For the toothpick method, 10 wooden toothpicks of normal size (6.5 cm) were inserted vertically into the soil surface of each pot (Figure 1). The tips of the toothpicks were lightly inserted into the soil surface (approximately 2–3 mm deep) before sunset. Overnight, the toothpicks were tilted or fell down by the earthworms foraging, and the next morning, the number of tilted or fallen toothpicks was counted and assigned a disturbance index (toothpick index) [31]. After counting, they were removed from the experimental units. This measurement was performed once a week, starting on the 15th day of the experiment, i.e., five times to assess legacy effects and six times to assess short-term effects after the new GBH/AI treatment. Two different levels of disturbance were used: a score of 0.5 for a titled toothpick and a score of 1 for a fallen toothpick. For each pot, the number of fallen toothpicks was multiplied by 1, and the number of inclined toothpicks was multiplied by 0.5. These numbers were summed and used as an index of aboveground activity for each pot.

Earthworm casts were collected twice a week. The casts on the soil surface were counted, collected, and stored in a paper bag. They were dried at 50 °C for 48 h and weighed. The cast collection was performed six times for legacy effects and eight times after herbicide treatment to determine short-term effects. The first collection of earthworm casts was on 23 October (day 21 of the experiment) because the earthworms needed some time to build their tunnels before they began excreting on the soil surface.

2.4.3. Litter Decomposition

Litter decomposition was assessed using the Tea Bag Index [42]. Therefore, plastic mesh bags of both green and rooibos tea were inserted on 2 October 2018, and 70 bags of Lipton rooibos tea (EAN 87 22700 18843 8) and 70 bags of Lipton green tea (EAN 87 22700 05552 5; Lipton Tea, Unilever Netherlands, Rotterdam, The Netherlands) were dried for one hour at 70 °C and weighed. The tea bags were then buried 8 cm deep in the center of the pots, with their labels kept above the surface. Each of the 70 pots received one bag of green tea and one bag of rooibos tea. The bags remained in the soil for 75 days, then they were removed and dried again for 48 h at 50 °C. After the removal of adhesive soil particles, they were weighed again. The litter stabilization factor (S) and the litter decomposition rate (k) were calculated based on the weight differences [42], assuming a hydrolysable fraction of 0.842 g g$^{-1}$ green tea and 0.552 g g$^{-1}$ rooibos tea (see also http://www.teatime4science.org, accessed on 19 July 2023).

2.4.4. Water Infiltration, Soil Temperature, Moisture, and Electrical Conductivity

On 11 December (28 days after herbicide application), a heavy rainfall event (20 mm) was simulated by pouring 1.5 l of tap water onto the soil surface of each pot using a watering can with a sprinkler. The water infiltration time was measured until the last water puddle was absorbed by the soil and used to calculate the water infiltration rate in mm s$^{-1}$. After this simulated rainfall, the leachate was collected in saucers placed underneath each pot and weighed to compare the water uptake capability of the pots.

Soil's moisture, temperature, and electrical conductivity were measured in all pots five times to assess legacy effects and five times after GBH/AI treatment using a portable time domain reflectometer (TDR) system (HD2, TRIME®-PICO 64/32, IMKO Micromodultechnik GmbH, Ettlingen, Germany). The TDR measuring fork was placed in the center of the pots for the measurements. The air temperature in the greenhouse was measured automatically by a datalogger installed in the greenhouse (MTV-model, HortiMax growing solutions, Hortisystems, Pulborough, UK). The mean air temperature in the greenhouse during the experiment was 18.9 °C in October, 17.5 °C in November, and 14.2 °C in December 2018.

*2.5. Termination of the Experiment*

At the end of the experiment (75 days after the start), the pots were flipped over, and the soil was carefully searched for earthworms (adult and juvenile) and earthworm cocoons. Juveniles and cocoons were then counted, and the reproduction index was calculated, with 0 points assigned if no juveniles or cocoons were found and 1 point if either juveniles or cocoons were found. The introduced earthworms were rinsed in water, dried, and weighed in order to compare their weight at the beginning with their weight at the end of the experiment; 17 of 210 adult earthworms could not be found anymore. It is not clear if they died or escaped. The earthworms that were still alive at the end of the experiment were released in the university's garden.

*2.6. Statistical Analysis*

Statistical analysis was carried out using the statistics program R version 3.6.1 for Windows [43]. The significance level was set to 5 % ($\alpha$ = 0.05).

A Generalized Linear Model (GLM) with a Poisson distribution was used to test the effects of GBH/AI and SOM on cumulative toothpick index and cumulative cast number. One model was calculated for legacy effects and another for short-term effects. A General Additive Model with Poisson distribution was conducted for the toothpick index and the cast number for short-term effects with the additional explanatory variable "day" (R-packages nlme and mgcv). Linear Models with a normal distribution were used for the variable cumulative cast weight (one model to analyze legacy effects and one after the herbicide application). Furthermore, an Additive Model with normal distribution for cast weight for the period after the herbicide application with the explanatory variable "day" was carried out. General Additive Models were run for toothpick index, cast number, and cast weight with additional soil parameter variables (soil's moisture, temperature, and electrical conductivity) for the period after herbicide application.

Linear Models with a normal distribution were run for the dependent variables: earthworm biomass (weight gain), earthworm number (at the end of the experiment), infiltration time, leachate, decomposition rate (k), and stabilization factor (S). A Generalized Linear Model with a binomial distribution was carried out for the variable reproduction. Soil temperature, moisture, and electrical conductivity were used as covariates in the models.

**3. Results**

*3.1. Legacy GBH/AI Effects*

Earthworm surface activity, as measured by toothpick index, cast numbers, and cast weight, was significantly affected by GBHs, with earthworms being more active on the surface after GBH application than after AIs or hand weeding (Table 3, Figure 2). Only the toothpick index was significantly affected by AIs, with lower activities than GBH but

slightly higher activities than after hand weeding. Glyphosate AIs were the only treatment that significantly interacted with SOM content, with effects more pronounced at low SOM content than at high SOM content (Table 3, Figure 2). Soil organic matter content as the main factor did not affect these earthworm activity parameters (Table 3).

**Table 3.** Statistical analysis of long-term legacy effects of glyphosate-based herbicides (GBHs), their respective active ingredients (AIs), soil organic matter contents (SOM), and their interactions on earthworm activity variables. P-values were obtained using Generalized Linear Models (GLMs) and Linear Models (LMs). Soil parameters were considered covariables. Bold values indicate significant influences.

| Parameter | GBH | AI | SOM | GBH × SOM | AI × SOM |
|---|---|---|---|---|---|
| Cumulative toothpick index (pot$^{-1}$) | **<0.001** | **<0.001** | 1.000 | 0.695 | **0.014** |
| Cumulative cast number (pot$^{-1}$) | **0.003** | 0.777 | 0.808 | 0.571 | 0.928 |
| Cumulative cast weight (g pot$^{-1}$) | **0.005** | 0.538 | 0.945 | 0.853 | 0.640 |
| Soil moisture (%) | **<0.001** | **0.002** | 0.228 | 0.624 | **<0.001** |
| Soil temperature (°C) | 0.075 | 0.246 | 0.530 | n.a. | n.a. |
| Soil's electrical conductivity (dS m$^{-1}$) | **0.032** | 0.682 | **0.001** | 0.309 | **0.005** |

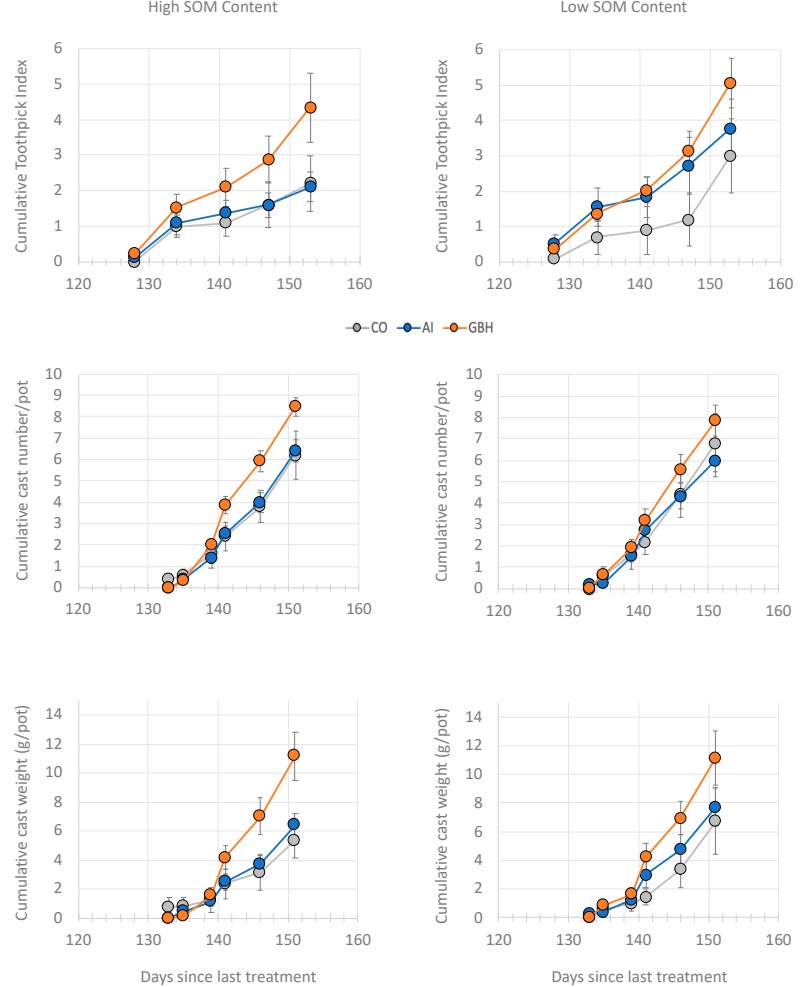

**Figure 2.** Long-term legacy effects of application of glyphosate-based herbicides (GBH), their active ingredients (AI), or hand weeding (CO) on earthworm surface activity at low (3.0%) and high (4.1%) soil organic matter (SOM) levels. GBH (Roundup PowerFlex, Touchdown Quattro, or Roundup LB Plus); AI (glyphosate potassium salt, diammonium salt, or isopropylamine salt). Means ± standard error (SE), *n* = 5.

When looking at the individual GBHs and AIs, it was found that the legacy effect of Touchdown Quattro significantly affected all three earthworm activity parameters, while Roundup LB Plus and Roundup PowerFlex each affected only two activity parameters (Table 4). Of the AIs, only glyphosate potassium salt (po), the AI in Roundup PowerFlex, showed a significant effect on toothpick index (Table 4). Only glyphosate potassium salt (po) interacted significantly with SOM (Table 4).

**Table 4.** Statistical analysis of long-term legacy effects of the glyphosate-based herbicides Touchdown Quattro (TQ), Roundup LB Plus (LB), and Roundup PowerFlex (PF) or their respective active ingredients, diammonium salt (am), isopropylamine salt (is), and potassium salt (po), SOM levels, and their interactions on earthworm activity variables. *p*-values were obtained using Generalized Linear Models (GLMs) and Linear Models (LMs). Bold values indicate significant influences.

| Parameter | TQ | LB | PF | am | Is | po | SOM | TQ × SOM | LB × SOM | PF × SOM | am × SOM | is × SOM | po × SOM |
|---|---|---|---|---|---|---|---|---|---|---|---|---|---|
| Cumul. toothpicks | **<0.001** | **<0.001** | **<0.001** | 0.423 | 0.844 | **<0.001** | 1.000 | 0.689 | 0.742 | **0.047** | 0.672 | 0.258 | **<0.001** |
| Cumul. cast no. | **<0.001** | 0.170 | **0.012** | 0.205 | 0.262 | 0.348 | 0.808 | 0.139 | 0.780 | 0.674 | 0.627 | 0.676 | 0.444 |
| Cumul. cast weight | **<0.001** | **0.024** | 0.313 | 0.128 | 0.982 | 0.985 | 0.945 | 0.970 | 0.388 | 0.174 | 0.746 | 0.638 | 0.191 |

Legacy effects of the previous GBH/AI treatments were also observed in soil parameters. The mean soil moisture was $20.9 \pm 1.1\%$ in CO, $24.1 \pm 1.2\%$ in AI, and $23.1 \pm 0.7\%$ in GBH at the beginning of the experiment and varied between 15.8% and 21.1% later in the experiment (Table 5). Soil moisture was higher in the pots under GBH or AI treatment than in the control; SOM had no effect on soil moisture (Table 5). The difference between treatment pots and control was significant in five of six cases: Roundup Powerflex ($p < 0.001$), Roundup LB Plus ($p < 0.001$), Touchdown Quattro ($p < 0.001$), diammonium salt ($p < 0.001$), and potassium salt ($p < 0.001$). Moreover, pots treated with Roundup LB Plus were moister than the corresponding AI pots treated with isopropylamine salt ($p < 0.001$). Pots treated with Touchdown Quattro or Roundup PowerFlex did not differ in soil moisture from those treated with the corresponding AI diammonium salt or potassium salt, respectively.

**Table 5.** Long-term legacy effects of previous treatments with glyphosate-based herbicides (GBH) or pure glyphosate active ingredients (AI) on soil parameters at low (3.0%) and high (4.1%) SOM levels. CO = control. Means $\pm$ SE, *n* = 5.

| Parameter | Low SOM | | | High SOM | | |
|---|---|---|---|---|---|---|
| | CO | GBH | AI | CO | GBH | AI |
| Soil moisture (%) | $17.2 \pm 0.8$ | $20.1 \pm 0.4$ | $21.1 \pm 0.5$ | $16.2 \pm 0.5$ | $20.8 \pm 0.6$ | $15.8 \pm 0.4$ |
| Soil temperature (°C) | $20.8 \pm 0.3$ | $20.8 \pm 0.1$ | $20.8 \pm 0.1$ | $20.8 \pm 0.3$ | $20.8 \pm 0.1$ | $20.8 \pm 0.1$ |
| Soil's electrical conductivity (dS m$^{-1}$) | $1.4 \pm 0.1$ | $1.5 \pm 0.0$ | $1.5 \pm 0.0$ | $1.2 \pm 0.0$ | $1.3 \pm 0.0$ | $1.1 \pm 0.0$ |

The mean soil temperature was $20.8 \pm 0.1$ °C and was not affected by the previous GBH/AI or SOM treatments (Table 5).

The mean electrical conductivity before treatment across all pots was $1.3 \pm 0.0$ dS m$^{-1}$; the treatments with GBHs and AIs had no effect on soil's electrical conductivity. SOM content significantly affected soil's electrical conductivity ($p = 0.001$): pots with low SOM had, on average, higher electrical conductivity than pots with high SOM content (Table 5).

### 3.2. Short-Term GBH/AI Effects on Earthworm Activity

Cumulative toothpick index and cast numbers were significantly affected by GBH treatments; only cast numbers were significantly affected by AIs (Table 6, Figure 3). SOM content only affected toothpick index but not cast number or cast weight (Table 6, Figure 3).

GBHs were also interactively affected by SOM; no interaction between AIs and SOM was observed for earthworm surface activity parameters in the short term (Table 6).

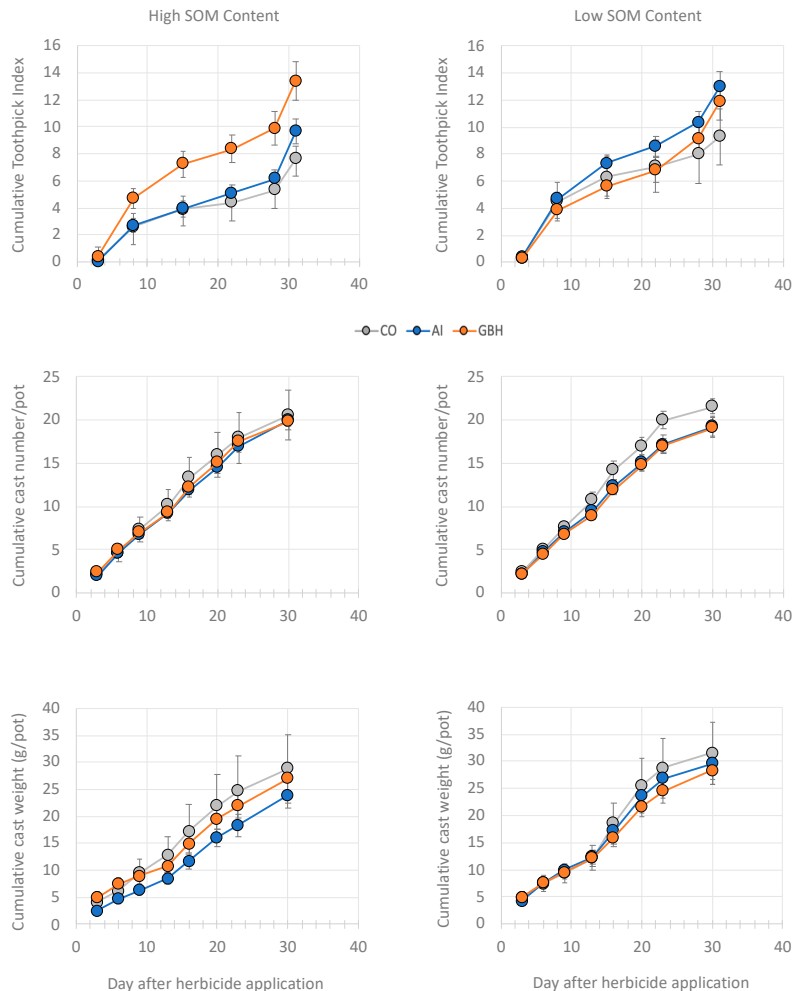

**Figure 3.** Short-term effects of glyphosate-based herbicides (GBHs), their active ingredients (AI), and mechanical weeding (CO) on surface activity of earthworms (toothpick index, cumulative cast numbers, and weight) at low (3.0%) and high (4.1%) soil organic matter (SOM) levels. GBH (Touchdown Quattro, Roundup PowerFlex, or Roundup LB Plus); AI (glyphosate potassium salt, diammonium salt, or isopropylamine salt). Means ± SE, *n* = 5.

Earthworm activity was additionally influenced by soil parameters. Soil moisture ($p < 0.001$) had a significant positive effect, and soil temperature ($p = 0.001$) had a significant negative effect on earthworm activity. Furthermore, there was a significant positive interaction between GBHs and soil moisture ($p < 0.001$) and AIs and soil moisture ($p = 0.036$).

Earthworm number ($2.76 \pm 0.57$ pot$^{-1}$), earthworm biomass ($13.0 \pm 3.09$ g pot$^{-1}$), and the reproduction index ($0.23 \pm 0.42$ pot$^{-1}$) were not affected by GBHs, AIs, or different SOM levels (Table 6).

Looking at the effects of each GBH or AI, it was found that Touchdown Quattro and Roundup LB Plus each affected two parameters of earthworm activity (toothpick index and cast numbers), while Roundup PowerFlex only affected toothpick index (Table 7). AI diammonium salt (am) significantly affected all three earthworm activity parameters (toothpick index, cast numbers, and weight); AI potassium salt (po) affected toothpick index only; and AI isopropylamine salt (is) did not significantly affect earthworm activity (Table 7). Soil organic matter only affected the toothpick index but not the other activity parameters. SOM interactively influenced the effect of Touchdown Quattro in terms of toothpicks and cast numbers, while SOM interactively influenced the effect of Roundup

LB Plus for toothpicks only (Table 7). No interactions were observed between SOM and Roundup PowerFlex or the AIs on earthworm activity (Table 7).

**Table 6.** Statistical analysis of short-term effects of glyphosate-based herbicides (GBHs), their active ingredients (AIs), soil organic matter content (SOM), and their interactions on earthworm activity parameters, earthworm number, biomass, and reproduction, water infiltration rate, and litter decomposition rate. *p*-values were obtained using Generalized Linear Models and Linear Models. Bold values indicate significant influences.

| Parameter | GBH | AI | SOM | GBH × SOM | AI × SOM |
|---|---|---|---|---|---|
| Cumulative toothpick index (pot$^{-1}$) | **<0.001** | 0.074 | **<0.001** | **<0.001** | 0.325 |
| Cumulative cast number (pot$^{-1}$) | **0.015** | **0.011** | 0.316 | 0.169 | 0.480 |
| Cumulative cast weight (g pot$^{-1}$) | 0.291 | 0.068 | 0.490 | 0.864 | 0.228 |
| Earthworm number end (pot$^{-1}$) | 0.925 | 0.849 | 0.747 | 0.643 | 0.455 |
| Weight change (g) | 0.762 | 0.410 | 0.414 | 0.849 | 0.736 |
| Reproduction index (ind. pot$^{-1}$) | 0.993 | 0.993 | 1.000 | 1.000 | 1.000 |
| Infiltration time (s L$^{-1}$) | **0.021** | 0.922 | **0.019** | 0.220 | 0.528 |
| Leachate amount (mL pot$^{-1}$) | **0.006** | 0.834 | **0.005** | 0.069 | 0.515 |
| Litter decomposition rate (k) | 0.649 | 0.538 | 0.379 | 0.641 | 0.294 |
| Litter stabilization factor (S) | 0.959 | 0.506 | 0.774 | 0.408 | 0.351 |
| Soil moisture (%) | **<0.001** | **0.048** | 0.435 | 0.475 | **<0.001** |
| Soil temperature (°C) | **0.020** | 0.142 | 0.392 | n.a. | n.a. |
| Soil's electrical conductivity (dS m$^{-1}$) | **0.002** | **0.036** | 0.143 | **0.160** | 0.065 |

**Table 7.** Statistical analysis of the glyphosate-based herbicides Touchdown Quattro (TQ), Roundup LB Plus (LB), and Roundup PowerFlex (PF), their active ingredients diammonium salt (am), isopropylamine salt (is), potassium salt (po), soil organic matter content (SOM), and their interactions on earthworm activity variables, earthworm number at the end of the experiment, biomass and reproduction, water infiltration rate, and litter decomposition rate. P-values were obtained using Generalized Linear Models (GLMs) and Linear Models (LMs). Bold values indicate significant influences.

| Parameter | TQ | LB | PF | am | is | po | SOM | TQ × SOM | LB × SOM | PF × SOM | am × SOM | is × SOM | po × SOM |
|---|---|---|---|---|---|---|---|---|---|---|---|---|---|
| Cumul. toothpick index | **<0.001** | **<0.001** | **<0.001** | **0.023** | 0.062 | **<0.001** | **<0.001** | **<0.001** | **<0.001** | 0.191 | 0.197 | 0.292 | 0.848 |
| Cumul. cast numbers | **<0.001** | **0.008** | 0.945 | **0.001** | 0.130 | 0.156 | 0.316 | **0.008** | 0.206 | 0.700 | 0.389 | 0.752 | 0.239 |
| Cumul. cast weight | 0.143 | 0.629 | 0.526 | **0.041** | 0.168 | 0.300 | 0.490 | 0.570 | 0.056 | **0.040** | 0.512 | 0.121 | 0.436 |
| Earthworm number end | 0.359 | 0.342 | 0.816 | 0.233 | 1.000 | 0.489 | 0.747 | 0.655 | 0.643 | 0.819 | 0.096 | 0.650 | 0.836 |
| Difference in weight | 0.536 | 0.661 | 0.580 | 0.253 | 0.906 | 0.463 | 0.414 | 0.494 | 0.953 | 0.273 | 0.349 | 0.327 | 0.426 |
| Earthworm reproduction | 0.993 | 0.994 | 0.993 | 0.993 | 0.994 | 0.993 | 1.000 | 1.000 | 0.998 | 1.000 | 0.998 | 0.998 | 1.000 |
| Infiltration time | 0.103 | **0.045** | **0.042** | 0.365 | 0.164 | 0.797 | **0.019** | 0.547 | 0.334 | 0.152 | 0.538 | 0.112 | 0.539 |
| Leachate amount | **0.002** | **0.022** | 0.055 | 0.173 | 0.389 | 0.965 | **0.005** | 0.176 | 0.101 | 0.073 | 0.246 | 0.410 | 0.658 |
| Decomposition rate (k) | 0.374 | 0.975 | **0.047** | 0.643 | 0.656 | 0.512 | 0.379 | 0.331 | 0.075 | 0.135 | 0.618 | 0.132 | 0.396 |
| Stabilization factor (S) | 0.651 | 0.649 | 0.896 | 0.668 | 0.687 | 0.097 | 0.774 | 0.882 | 0.778 | 0.108 | 0.616 | 0.624 | 0.189 |

### 3.3. Water Infiltration and Leachate

Water infiltration rate and leachate were significantly affected by GBHs and SOM but not by AIs; there was no interaction between GBHs or AIs and SOM (Table 6; Figure 4). The mean infiltration was $3.3 \pm 0.4$ mm s$^{-1}$ in pots with high SOM content and $1.0 \pm 0.2$ mm s$^{-1}$ in pots with low SOM content. Looking at individual GBHs or AIs, infiltration was also significantly affected by Roundup LB Plus and Roundup PowerFlex, but not by Touchdown Quattro or any of the AIs tested (Table 7).

Leachate amount was significantly affected by GBHs and SOMs but not by AIs (Table 7, Figure 4). The addition of 1.5 l pot$^{-1}$ of water caused a mean leachate of $723 \pm 26$ mL in

low SOM pots and a mean leachate of $550 \pm 36$ mL in high SOM pots (Figure 4). Looking at individual GBHs or AIs, leachate was significantly affected by Touchdown Quattro, Roundup LB Plus, and marginally by Roundup PowerFlex, but not by any of the AIs tested (Table 7).

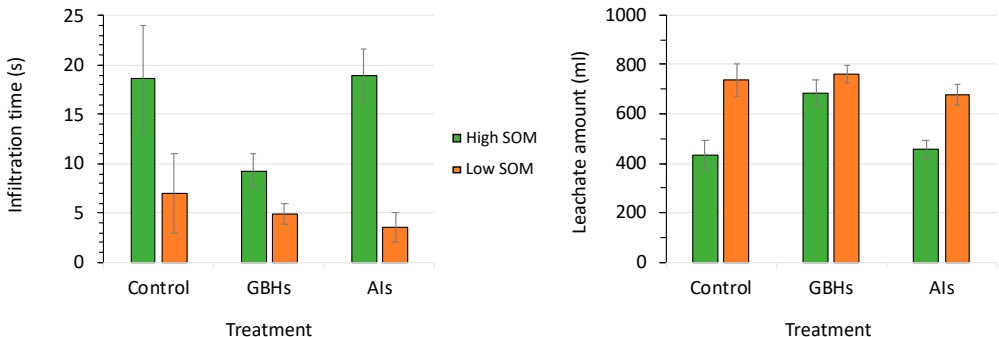

**Figure 4.** Infiltration time and leachate amount under hand weeding (control) or treatment with glyphosate-based herbicides (GBHs) or respective active ingredients (AIs) under low (3.0%) or high (4.1%) soil organic matter content (SOM). Means $\pm$ SE, $n = 6$.

### 3.4. Litter Decomposition and Soil Parameters

The decomposition rate across treatments was $0.0027 \pm 0.0028$ and the stabilization factor across treatments was $0.030 \pm 0.021$, but neither GBHs, AIs, nor SOMs affected the litter decomposition rate or the stabilization factor (Table 6).

Soil moisture and electrical conductivity were significantly affected by GBHs and AIs, with an interaction between AI $\times$ SOM; SOM alone had no significant effect on soil moisture (Table 6). Soil temperature was affected only by GBH but not by AI or SOM (Table 6).

When considering individual GBHs and AIs, soil moisture was significantly higher compared to control under Touchdown Quattro ($p < 0.001$), Roundup LB Plus ($p < 0.001$), diammonium salt ($p = 0.011$), and potassium salt ($p = 0.021$). Moreover, soil moisture in pots with high SOM content was significantly positively influenced by GBHs ($p < 0.001$) (Table 8). In pots with low SOM content, soil moisture was significantly higher under GBH and AI applications than in the control (both $p < 0.001$; Table 8).

**Table 8.** Soil parameters under hand weeding (CO) or treatment with glyphosate-based herbicides (GBH) or their active ingredients (AI) at low (3.0%) or high (4.1%) soil organic matter contents (SOM). Means $\pm$ SE, $n = 6$.

| Parameter/Period after Treatment | Low SOM | | | High SOM | | |
|---|---|---|---|---|---|---|
| | CO | GBH | AI | CO | GBH | AI |
| Soil moisture (%) | $20.3 \pm 1.0$ | $23.7 \pm 0.5$ | $23.7 \pm 0.5$ | $19.5 \pm 1.0$ | $22.0 \pm 0.6$ | $18.9 \pm 0.6$ |
| Soil temperature (°C) | $16.3 \pm 0.4$ | $17.1 \pm 0.2$ | $16.5 \pm 0.2$ | $16.3 \pm 0.3$ | $17.0 \pm 0.2$ | $17.0 \pm 0.3$ |
| Soil's electrical conductivity (dS m$^{-1}$) | $1.1 \pm 0.1$ | $1.3 \pm 0.0$ | $1.3 \pm 0.0$ | $1.0 \pm 0.0$ | $1.0 \pm 0.0$ | $1.0 \pm 0.0$ |

Soil's electrical conductivity was significantly higher across SOM in pots treated with GBHs ($p = 0.002$) and AIs ($p = 0.036$) compared to control pots (Table 8).

### 4. Discussion

This is one of the first studies to investigate both the long-term legacy and short-term effects of different glyphosate AIs and their GBH formulations on earthworms and soil functions at different soil organic matter (SOM) levels. We found that both the long-term and short-term effects of GBHs on earthworm activity, water infiltration, and water leaching

were higher than those of AIs. Although SOM itself influenced the parameters studied, few interactions were observed between GBHs, AIs, and SOM.

### 4.1. Long-Term Legacy Effects

Earthworm activity was still increased when GBHs were applied almost 4 months ago, while no such legacy effect was observed after the application of AIs or after hand weeding. We explain this mainly by the higher soil moisture under GBH treatment, as higher soil moisture often leads to higher earthworm activity [44–46]. This higher soil moisture under GBH was surprising because the soil was uniformly stored after the previous GBH/AI application and the soil moisture levels were not significantly different before the start of the current experiment. The higher soil moisture after GBH application is consistent with another study examining the same GBHs/AIs and hand weeding control [13]. One interpretation of our current results could be that co-formulants were added to GBHs and/or the earthworms involved influenced soil hydrology. More detailed studies are necessary to better understand these relationships between GBHs, earthworms, and soil moisture. An 8-week study showed that GBH treatment (Roundup Alphee and Roundup Speed) resulted in increased soil moisture as physiologically active transpiring plants were killed; however, the current study showed adverse effects on earthworm surface activity [28]. In the current experiment, plants in the control treatment were uprooted to simulate the effect of mechanical weeding (e.g., harrowing), which may have affected soil moisture.

Higher earthworm activity under GBH compared to AI or hand weeding was not expected, as we assumed that co-formulants added to AIs would be more detrimental than AIs alone. Since the ingredients are not disclosed, the reason for this remains unclear. However, a recent study shows that a co-formulant (alkylpolyglycoside APG) can indeed stimulate earthworms [17]. Therefore, we interpret the higher surface activity as a stress response to the substances in the GBHs but not to AI alone or hand weeding. In contrast, earthworms under AI or hand weeding may have lived in deeper soil horizons and avoided foraging and casting at the surface. This could also be an indication of the higher water infiltration in pots with AI or mechanical weeding (see below). However, to investigate this further, all the ingredients of the GBHs studied need to be known.

In any case, our observations of legacy effects showed that the application of GBH (and, to a lesser extent, AI) still had effects on earthworms 4 months after application. This also indicates that ecologically relevant effects on earthworms last much longer than the 6.5-day $DT_{50}$ half-life reported for glyphosate [47]. Of course, the half-life also depends on soil conditions such as pH, soil texture, soil moisture, soil temperature, and microbial activity [1,48]. Moreover, legacy effects of glyphosate residues in soil have been shown to alter plant biochemistry and may affect plant interactions with herbivores and mutualistic organisms, as well as impair plant resistance and attraction to beneficial insects [40,49,50].

### 4.2. Short-Term Effects of GBHs versus AIs

We hypothesized that GBHs would have a stronger effect on the parameters studied than individual AIs or hand weeding. Indeed, GBHs affected two parameters of earthworm activity (toothpicks and cast numbers), while AIs significantly affected only cast numbers (but marginally also affected the toothpick index and cast mass). Surprisingly, however, earthworm surface activity (based mainly on toothpick index) increased after the application of GBH compared to AIs and hand weeding. Moreover, this difference between GBHs and AIs was particularly the case at higher SOM levels (significant GBH × SOM interaction), suggesting that GBHs could have been bound to SOM and been effective longer. The toothpick index measures the aboveground mobility of earthworms when they come to the soil surface to forage, and casting activity measures their excretory activity at the soil surface. Thus, our results suggest that earthworms were more active or stressed in foraging at the soil surface after GBH application. Exposure of *Eisenia fetida* to GBH (Roundup Ready-to-Use III) increased the rate of movement in a one-week experiment [51].

This is in contrast to some short-term studies running up to six weeks, which suggest that sublethal doses of herbicides can stimulate earthworms as a result of increased nutrient availability in the soil [48,52,53]. Higher SOM content leads to higher soil moisture and earthworm activity under GBH, but it is unclear why no interaction between AI and SOM was observed. The fact that SOM only affected the toothpick index suggests different SOM-related soil properties that favored earthworm behavior at the soil surface. Here, soils with higher SOM also had 54% higher phosphorous and 67% higher potassium contents than soils with low SOM. Intrinsic characteristics of earthworms, such as initial body mass, have also been shown to affect their response to herbicides [53].

Earthworms (*E. fetida)* living in soil amended with the GBHs (Roundup Ready-to-Use III and Roundup Super Concentrate) did not lose body mass and survived a stress test as well as worms living in uncontaminated soil for 40 days [54]. The authors suggest that nitrates and phosphates in the GBHs offset the toxic effects of the AIs by stimulating microbial growth and accelerating glyphosate degradation [54].

Neither earthworm numbers, biomass, nor reproduction index were affected by GBHs, AIs, or SOM levels in our study. Field studies also showed that no effects on earthworm biomass were observed at the recommended field application rates after six weeks [39], but stimulation of soil microorganisms was reported [16].

Other studies that ran for several weeks found both higher reproduction of *Eisenia* species in glyphosate-treated soils [48,52,55] and lower reproduction of *Eisenia* and *Lumbricus* species in soil treated with either AIs or GBHs [28,56–58]. The current experiment used an anecic earthworm species that has a longer reproduction cycle than epigeic *Eisenia* species; most of the previous studies also used higher herbicide doses. Other studies have shown that GBHs (Roundup Alphée) also affected biochemical and physiological biomarkers (enzymatic activities and antioxidant defense) of different earthworm species (*Alma millsoni*, *Eudrilus eugeniae*, and *Libyodrilus violaceus*) [59,60].

Few studies have examined the effects of GBH/AI on earthworms at different SOM levels [13]. Since the herbicide dose used in the current experiment was moderate and soil conditions were suitable for earthworms, our results show that GBH has stronger effects than AIs, considering that high activity is positive. Similarly, earthworms in soils treated with GBHs were found to lose no biomass and survive a stress test, whereas earthworms in soils amended with pure glyphosate lost weight and survived a stress test for a significantly shorter time than individuals in uncontaminated soils [54]. A recent study suggests that co-formulants (APGs) may also stimulate earthworm reproduction [17]. Another possibility for the difference between GBHs and AIs could be that co-formulants reduce [53] or stimulate soil microorganisms [16], thus indirectly affecting earthworms [25].

In general, GBHs in our study increased earthworm activity (toothpicks, cast numbers, and weight) to a greater extent than the corresponding AIs. The results of other studies are different, as often GBHs are more toxic than AIs [21,61]. Surprisingly, according to the safety data sheets of the three GBHs studied here, only Roundup LB Plus was tested on earthworms, whereas the other GBHs, Touchdown Quattro and Roundup PowerPlus, were not tested on earthworms. Moreover, ecological risk assessments are mainly conducted on the relatively resistant earthworm species *E. fetida/andrei,* which is also generally not found in arable land [62]. This underscores the need for ecological risk assessments for GBHs, including co-formulants, and more realistic testing of various non-target organisms in agricultural fields by independent regulatory agencies.

Since the ingredients of GBHs are not known, we can only speculate that: (i) a co-formulant stimulated earthworms; (ii) there are indirect effects of GBHs on soil moisture; or (iii) both factors were responsible for this. Our current results contrast with a previous study that showed reduced earthworm casting activity under treatment with the same GBHs and AIs [13]. We hypothesize that the differences were due to the soil temperature of 24 °C in the earlier experiment, which was less favorable for earthworms, whereas we had a more favorable 19 °C in the current experiment. Thus, interactions with environmental factors seem to be crucial for the manifestation of the effects of GBHs or AIs. Indeed, it has

been shown that environmental characteristics such as soil temperature can have a major impact on earthworm responses to GBH and AI applications [17,53] and other non-target organisms such as amphibians [63,64]. Several other studies found reduced earthworm activity under glyphosate treatments [28,65]. Other studies indicated that glyphosate AI and GBH Roundup had no effect on *E. fetida* at recommended field doses [55,66].

In the current study, no effect of GBH/AI treatments on litter decomposition rate or stabilization factor was observed, which may be due to the short experimental period, although litter clearly disappeared from the litter bags during this time. Other studies also found little effect of GBH/AI treatments on litter decomposition [28,65,67], suggesting that decomposer organisms are rather irresponsive to glyphosate AIs, GBHs, or SOM contents.

Comparisons between different study results are difficult due to the variety of glyphosate AIs and GBHs available. In addition, the different earthworm species used respond differently. The anecic species used in our experiment live in permanent burrows in deep soil layers and come to the soil surface to feed or deposit their castings [46]. They differ from epigeic *Eisenia* species commonly used in environmental risk assessments [68]. Only one of the other studies used the same herbicides, the same rather low herbicide doses, and the same earthworm species as in the current experiment [13]. Additionally, depending on the type of GBH, the amount of AI and the amount and composition of adjuvants vary, and some herbicide formulations are more toxic to earthworms than others [62]. Hence, GBHs are black boxes because the ingredients are not known, and even GBHs of the same brand may have different ingredients in different countries [69].

### 4.3. Water Infiltration and Leaching

Water infiltration after a simulated 20-millimeter extreme rainfall event and the amount of leachate from experimental pots were significantly affected by the SOM content and by the GBH application, but not by the AI application. Infiltration time was generally longer under high SOM than under low SOM. Moreover, infiltration was mainly affected by AI/GBH at high SOM. Infiltration time was similar between AIs and hand weeding but twice as long, and the amount of leachate was more than 30% lower in pots treated with AIs and hand weeding compared to pots treated with GBH at high SOM content. Glyphosate residues in leachate were not analyzed, but assuming leachates had the same residue concentrations, more glyphosate residues would reach groundwater in GBH-treated soil than in AI-treated soils.

It is unclear why the different GBHs showed different effects; for instance, Roundup LB Plus had the shortest infiltration time of all GBHs tested, followed by Touchdown Quattro, while pots treated with Touchdown Quattro had the highest amount of leached water. More detailed analyses of the causes cannot be performed until the complete ingredients are known.

The effect of SOM on infiltration time and leachate volume can be attributed to the positive correlation between SOM content and water holding capacity [70]. The lower water holding capacity of the pots treated with GBHs can be explained by the higher soil moisture prior to the heavy rain event. In particular, soil moisture was significantly higher in the pots with high SOM when combined with GBHs. The higher soil moisture resulted in higher earthworm movement activity in the pots treated with GBHs, which increased water infiltration belowground [71].

### 5. Conclusions

We observed that mainly GBH and, to a lesser extent, AI treatments affected earthworm activity. SOM content had little effect on earthworm activity, and only an interaction between SOM and AI was observed. This suggests more stressed earthworms at the surface, likely consuming less food than indicated by casting. The effects of GBHs or AIs on earthworm behavior were likely due to a change in soil moisture, which was higher in pots treated with GBHs than in AI treatments. The effects of GBH/AIs on earthworms affected water infiltration rate and leachate, which were higher in GBH pots. It can be

concluded that the effects of commercial GBHs differed from those of pure AIs; moreover, the three GBHs and AIs also differed in their individual impacts. This is important because environmental risk assessments largely focus on pure AIs rather than formulated GBHs, even though AIs represent only a fraction of GBHs. In addition, the non-disclosure of all environmentally relevant ingredients hinders rigorous analysis of the ecological impacts of the world's most widely used herbicides. Recent research shows that petroleum appears to be a common ingredient or contaminant in commercial pesticides [72,73], providing an additional explanation for the difference between AIs and GBHs. Our study highlights the need for more realistic risk assessments for the approval of GBHs and AIs for non-target soil organisms that also include biotic and abiotic soil parameters.

**Supplementary Materials:** The following supporting information can be downloaded at: https://www.mdpi.com/article/10.3390/soilsystems7030066/s1. Table S1: DATA_earthworm_casting_MS_Brandmaier.xlsx; Table S2: DATA_earthworm_toothpicks_MS_Brandmaier.xlsx; Table S3: DATA_soil_parameters_MS_Brandmaier.xlsx; Table S4: DATA_infiltration_MS_Brandmaier.xlsx.

**Author Contributions:** Conceptualization, V.B., A.A., E.G., A.S. and J.G.Z.; methodology, V.B., A.A., E.G., E.T., M.M., S.K., A.S. and J.G.Z.; validation, J.G.Z., E.G. and A.S.; statistical analysis, V.B. and F.L.; investigation, V.B. and A.A.; resources, E.G., M.M. and S.K.; data curation, V.B., E.G. and J.G.Z.; writing—original draft preparation, V.B. and J.G.Z.; writing—review and editing, all authors; visualization, V.B. and J.G.Z.; supervision, A.S., E.G. and J.G.Z.; project administration, A.S., M.M., E.G. and J.G.Z.; funding acquisition, A.S. and J.G.Z. All authors have read and agreed to the published version of the manuscript.

**Funding:** This research was funded by project no. 97öu3 of the Austria-Hungary action of the Osztrák-Magyar Akció Alapítvány (OMAA) and the Austrian Agency for International Cooperation in Education and Research (OED) granted to A.S. and J.G.Z. The funding body had no role in the design of the study, the collection, analysis, and interpretation of data, or in writing the manuscript.

**Institutional Review Board Statement:** No ethical review or approval was necessary for this study because only invertebrates were studied.

**Informed Consent Statement:** Not applicable.

**Data Availability Statement:** All raw data is provided in the Supplementary Material.

**Acknowledgments:** We are grateful to Yoko Muraoka, Ricarda Schmidt, Georg Mayrpeter, and Johannes Altmanninger for various kinds of help during the experimental and writing phases. Thanks to the BOKU Experimental Farm for providing the soil.

**Conflicts of Interest:** The authors declare no conflict of interest.

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
