# Peer review of "Glyphosate-Based Herbicide Formulations with Greater Impact on Earthworms and Water Infiltration than Pure Glyphosate"

_soilsystems, doi:10.3390/soilsystems7030066_

Round 1

Reviewer 1 Report

The manuscript results are of interest to researchers in environmental sciences, but need make some modifications before to consider to be accepted for publication.

                     Discuss the results with other studies where Eisenia fetida and other organisms have been tested.

                     The authors mention the effects of GBH or AI on earthworms in the short and long term. What do they refer to as long-term with respect to other studies? Indicate the time ranges in which these studies has been carried out.

                     The earthworm activity assay is subjective and qualitative; there is no quantitative support. Lack of support with biochemical analysis or weight and size of the aearthworms

                     Review the days the experiment lasted in the methodology mentions 75 and 76 days (line 136).

                     In Figure 2, why do some graphs begin before 130 days and others after these days?

                     Table 3 and Figure 2 should be described in greater detail and separately in the results section.

                     In Figure 3, it is suggested to place the days from 1 to 28 and adjust the scale on the X-axis to appreciate the 28 days better.

Additional commentaries:

Line 292, 324: Place the electric word without abbreviation

Line 404: Remove the letter T that is before the word Soil

The authors can minor editing of English language required

Reviewer 2 Report

The manuscript entitled "Glyphosate-based Herbicide Formulations with Greater Impact on Earthworms and Water Infiltration than Pure Glyphosate" is generally a well-written, well-organized and illustrated. It presents the results of original research and makes a valuable contribution to knowledge and understanding of the impacts of glyphosate-based herbicides on earthworms and water infiltration. The results and discussion are presented very concisely.

Specific comments:

-Title: This manuscript overlaps with a previously published paper entitled "Effects of glyphosate-based herbicides and their active ingredients on earthworms, water infiltration and glyphosate leaching are influenced by soil properties; https://doi.org/10.1186/s12302-021-00492-0" by the same authors. It seems like the only difference between the previous study (published in 2021) and the current study is the time of glyphosate application! In order to understand this subject, it is important to provide an explanation.

-Introduction: In the introduction, you may discuss the toxicological effects of Glyphosate on different animals and in particular earthworms. In addition, it is important to discuss the environmental fate of glyphosate in soil.

-Materials and Methods: lines 88-90: Replications should be mentioned.

-Materials and Methods lines 91-93: Is the only difference between the previous studies and this one the time of herbicide application?!

- Materials and Methods lines 98-100: Describe the history of herbicide application in the fields where you collected the tap soil samples.

Materials and Methods table 2: In order to convert active ingredients in hectares into active ingredients for pots, what was the statistical basis?

- Materials and Methods line 138: In light of the use of earthworms in this research, explain the ethics of animal experiments.

- Materials and Methods line 158: In what stage of plant growth is herbicide applied?

- Materials and Methods lines 254-255: Which package of R software did you use?

Acceptable.

Reviewer 3 Report

The manuscript titled “Glyphosate-based Herbicide Formulations with Greater Impact on Earthworms and Water Infiltration than Pure Glyphosate” is a short-term microcosm experiment comparing the effects of glyphosate-based herbicides and glyphosate active ingredients on earthworm activity and soil properties. The topic is timely and important. The study is well developed and the manuscript is well-written. I have just some remarks and suggestions below.

Abstract:

I would mention in the Abstract that it is a microcosm/pot experiment.

Introduction:

Lines 75-85: I would also mention here, that it is a greenhouse pot experiment.

Line 86: As I understood, you did not measure glyphosate leaching but the amount of leached water.

Materials and Methods:

Lines 113-120: “Three-factorial design” is not accurate. Actually, there were only two factors: SOM and herbicide treatment. SOM had two levels and herbicides had 7 levels. However, the herbicide levels were not independent from each other, as TQ contained am, but it did not contain po and is. PF contained po but did not contain am and is etc.

It was also not clear how did you build up the models with these factors. Were GBH and AI levels in two different models? I can imagine this model with two factors. How can you take into one model the GBH treated and AI treated data if you treat these two types of treatments as two factors?

Line 129: You used dried soil for the experiment. I missed the data about the moisture level when you started the experiment. How did you standardize the soil moisture at the beginning?

Figure 1 label: I would explain the abbreviation TDR.

Lines 158-165: These sentences have strange structure. In line 161 the word “or” cannot finish the sentence.

Line 165: Mustard plants were removed with their roots?

Line 184: If you applied a type of herbicide, was it applied in such soil, where the legacy effect would be the same? I mean, if you applied PF in a pot, did this pot contain such soil which was applied with PF before? Then the control soil was not applied with any herbicides before?

Lines 188: You measured the plant growth but you did not show any results of them but I am really interested in the effects of treatments on plants.

Line 248: How did you calculate reproduction rate?

Line 250-251: Just a curious question: what happened to the earthworms after the experiment? As drying means in this situation just drying with paper towels, they did not die- as I think.

Results

Figures: n=6? I do not understand this 6. The replicate of the treatments was 5. For instance, in Figure 2, one point in a plot related to AI or GBH contained 3 treatments with 5 replicates, so it means these points contained 3×5=15 data, and for control, it should be 5.

Tables: For such models, it is not enough adding only the p-values. What about the R values, the coefficients?

Lines 390-391: I missed the results about decomposition rate, however, the duration of the experiment was not so long for a decomposition study.

Table 8: TSoil?

Discussion:

In control pots, you removed the plants. I think it would affect the soil moisture. Without plant cover, it can dry out faster than with dead plant material cover. And you compared everything to the control.

Line 420-421: That is why I am interested in the rewetting method of the soils.

Line 431: What is APG?

Line 456: You mention Eisenia fetida here first time, write the whole name.

Line 529: reference

Round 2

Reviewer 1 Report

I reviewed the response letter and I suggest that in future studies you consider quantitative assays and include more robust biochemical or physiological biomarkers to evaluate the effects of GBH on Eisenia fetida or other organisms they use. Likewise, you should specify the times at which the experiment was performed and not just mention short or long in the wording.

The authors can minor editing of English language required

Reviewer 2 Report

In my opinion, this manuscript could be accepted since all my comments have been addressed by authors.

Reviewer 3 Report

Thank you for the responses of my comments and suggestions!

However, the answers generated further comments.

First, it would be more polite from the authors if they add the text line numbers where they corrected their manuscript. I also added line numbers so you could find immediatelly what I was writing about. I suggest to do this in the future.

Second, you added the starting moisture data of the soils but they were not the same among the treatments and I think it would influence the later soil moisture data. In addition, the plants were not removed but uprooted. It also influences the soil moisture.  I would add a discussion part about this. If this sentence is true: "the soil moisture levels were similar before the start of the current experiment" (line 484), then why are the moisture differences in Table 5 interesting? I mean, if the ~4 % moisture range at the beginning is "similar", than later, in Table 5, this ~6 % moisture range is "different"?

Why do you call reproduction rate as rate? 0 and 1 do not express any rates but presence and absence. It is rather an index. 
